# Modern Paediatric Emergency Department: Potential Improvements in Light of New Evidence

**DOI:** 10.3390/children10040741

**Published:** 2023-04-17

**Authors:** Roman Kula, Stanislav Popela, Jozef Klučka, Daniela Charwátová, Jana Djakow, Petr Štourač

**Affiliations:** 1Department of Paediatric Anaesthesiology and Intensive Care Medicine, University Hospital Brno and Faculty of Medicine, Masaryk University, Kamenice 5, 625 00 Brno, Czech Republic; roman-kula@hotmail.com (R.K.);; 2Department of Physiology, Faculty of Medicine, Masaryk University, Kamenice 5, 625 00 Brno, Czech Republic; 3Emergency Department, University Hospital Olomouc and Faculty of Medicine, Palacký University, I.P. Pavlova 185/6, 779 00 Olomouc, Czech Republic; 4Emergency Medical Service of the South Moravian Region, Kamenice 798, 625 00 Brno, Czech Republic; 5Department of Simulation Medicine, Faculty of Medicine, Masaryk University, Kamenice 5, 625 00 Brno, Czech Republic; 6Department of Surgery, Vyškov Hospital, Purkyňova 235/36, 682 01 Vyškov, Czech Republic; 7Paediatric Intensive Care Unit, NH Hospital Inc., 268 01 Hořovice, Czech Republic

**Keywords:** emergency, triage, guidelines, checklist, ultrasound

## Abstract

The increasing attendance of paediatric emergency departments has become a serious health issue. To reduce an elevated burden of medical errors, inevitably caused by a high level of stress exerted on emergency physicians, we propose potential areas for improvement in regular paediatric emergency departments. In an effort to guarantee the demanded quality of care to all incoming patients, the workflow in paediatric emergency departments should be sufficiently optimised. The key component remains to implement one of the validated paediatric triage systems upon the patient’s arrival at the emergency department and fast-tracking patients with a low level of risk according to the triage system. To ensure the patient’s safety, emergency physicians should follow issued guidelines. Cognitive aids, such as well-designed checklists, posters or flow charts, generally improve physicians’ adherence to guidelines and should be available in every paediatric emergency department. To sharpen diagnostic accuracy, the use of ultrasound in a paediatric emergency department, according to ultrasound protocols, should be targeted to answer specific clinical questions. Combining all mentioned improvements might reduce the number of errors linked to overcrowding. The review serves not only as a blueprint for modernising paediatric emergency departments but also as a bin of useful literature which can be suitable in the paediatric emergency field.

## 1. Introduction

Recently published papers have outlined the increasing overcrowding in both adult [1] and paediatric [2] emergency departments. High levels of stress exerted on emergency physicians, together with an environment full of multitasking and interruption, inevitably lead to an elevated rate of task errors [3]. In light of patient safety, more attention is now focused on the methods guaranteeing an equal quality of care to all incoming patients, implementing structural thinking to buy a physician’s mental space for important decisions and giving more accuracy to discriminate patient’s diagnosis.

The present review thus brings possible improvements in the three above-mentioned domains. The aim is to make the improvements easily incorporable into daily clinical routines. We have also highlighted the crucial publications which served us to compile the ideological framework for the proposed improvements.

## 2. How to Optimise Workflow in Paediatric Emergency Departments?

Overcrowding in paediatric emergency departments remains an important public health concern. In contrast to adult emergency departments, paediatric patients do not have higher odds of hospital admission or mortality after being discharged from the overcrowded emergency department [4]. However, overcrowding may negatively influence the quality of care, e.g., delays in antibiotic administration for febrile neonates, analgesia for sickle cell crises, or treatment of acute asthmatic exacerbation [5,6,7]. Moreover, children coming to crowded paediatric emergency departments also have a higher likelihood of being admitted [4,8]. It, therefore, remains essential to understand the causes of overcrowding.

Before mitigating the causes, the level of overcrowding must be correctly estimated. The investigators traditionally measure retrospective indicators, such as waiting time for examination by a physician, total length of stay in the emergency department or proportion of patients leaving the emergency department without being seen by a physician. In addition to these unidimensional indicators, two multi-dimensional scores (PEDOCS [9] and SOTU-PED [10]) were designed to obtain data from real-time paediatric emergency department operations and inform staff and administrators if crowding occurs. Both scores were critically evaluated in the recently published review [11] and found to be comparably accurate. In PEDOCS, the score is calculated according to Equation (1), and the scale ranges from 0 to 200 (0, not busy; 40, busy; 80, extremely busy but not overcrowded; 120, overcrowded; 160, severely overcrowded; 200, dangerously overcrowded).
PEDOCS = 33.3 × 0.11 + 0.07 × *(patients in the waiting room)* + 0.04 × *(total registered patients)*(1)

SOTU-PED is a linear model, defined by Equation (2), to predict global hourly crowding perception on a 10-level Likert scale. Perception of overcrowding among healthcare professionals occurred within the value greater than 5 and corresponded with a SOTU-PED of 2 and higher.
SOTU-PED = 0.764 + 0.49 Census-H24 (number of admissions in the past 24 h) + 0.496 Occ-Rate (occupancy rate) + 0.302 1-year infant (number of patients)(2)

Once the level of overcrowding is identified, the input-throughput-output model of patient flow in the emergency department might help to find gaps for improvement [12]. The most promising way to reduce the burden of paediatric patients on the input side remains to divert non-urgent patients at triage (i.e., levels 4 and 5 in all routinely used triage systems) to nearby alternative locations [12,13,14]. These units, so-called fast tracks, are urgent care centres or retail clinics, usually staffed by experienced practitioners or physician assistants, respectively. In addition to their application in fast tracks, triage systems generally facilitate the prioritisation of patients by assigning them to one of the predefined levels (usually five in total) of urgency with the dedicated maximum possible waiting time. Widely used triage systems with available paediatric versions are the Australasian Triage Scale (ATS), Canadian Triage and Acuity Scale (CTAS), Emergency Severity Index (ESI), Manchester Triage Scale (MTS), and South African Triage Scale (SATS). These triage systems are comparable (Table 1) and share a standardised format: deploy a 5-level classification scheme and set targets for timeliness to physician contact per triage level [15]. None of the above triage systems emerges as superior, and similar performance trends and weaknesses are common to all systems [15,16].

Triage systems usually rely on an experienced triage nurse to undertake triage [17]. Employing primary healthcare professionals (i.e., general or nurse practitioners) may be a useful extension of the triage team [18]. Discussion about replacing a triage nurse with a physician has no evidence to suggest that physicians are any better or more cost-effective at triage than experienced nurses [17,19]. Triage systems may work more smoothly when combined with artificial intelligence. Based on the previously collected data, artificial intelligence learns to predict the value of any targeted parameter with a certain level of accuracy. For example, timeliness to physician contact per triage level could be accompanied by the prediction of real waiting time based on the ongoing level of crowding [20]. This information is essential to make more responsive and proactive actions (i.e., asking the doctor-on-call to be on duty or deploying doctors from other departments) if a long waiting time is anticipated. Artificial intelligence may match patients to triage levels even more accurately than emergency specialists themselves [21]. Further possibilities for the implementation of artificial intelligence in the field of emergency medicine were systematically reviewed by Boonstra and Laven [22].

Worster et al. highlighted the importance of triage education since they showed that after 3 h of triage training, general nurses were able to match experienced nurses in the use of the triage system [23]. The majority of the attainable possibilities for paediatric triage education were recently summarised in the integrative review [24]. Among the wide variety of strategies, such as standardised educational programs, patient simulations followed by structured debriefing, and computerised paediatric scenarios or lectures, the patients’ simulations are the most reliable not only in gaining but also in sustaining triage skills [25,26,27]. Retraining with a certain frequency is also crucial since it was shown that participants of the most well-represented Emergency Triage Assessment and Treatment course, which includes both didactic and hands-on approaches, experienced a decline in triage skills over time [28].

An alternative to the triage system is the ‘see and treat’ model, which has been available in the UK since 2004 [29]. The model is set to treat less severe patients as soon as they arrive at the emergency department. One team of clinicians is dedicated to ‘see and treat’: they assess incoming patients and immediately treat and discharge those with minor complaints. Simultaneously, another team deals with more serious cases, triaged after the initial assessment. Since its introduction, the ‘see and treat’ model has been broadly used in the UK [30] and is probably responsible for the largest overall reduction in waiting times. The clinicians of the first contact with minor illness are physicians or, more often, nurses. As was proved by Sakr et al. [31], nurses with at least 4 years of experience working in the emergency department can treat patients with minor injuries equally well as junior doctors. This finding highlights the importance of taking nurses as reliable partners throughout providing paediatric emergency care. It is also necessary to habilitate nurses with an adequate training program and procedural competencies. The trained nurse can, for example, successfully place peripheral intravenous catheters under ultrasound control if intravenous access is recognised to be difficult to secure [32].

The pandemic of COVID-19 has brought a strong involvement of virtual meetings to everyday life. With regard to available online technologies, Reid et al. [33] examined the feasibility, utilisation rate and satisfaction of virtual care as an adjunct to in-person emergency care. The authors adapted a secure encrypted video platform (Zoom for Healthcare™). Prior to meeting an emergency physician online, the patient went through an online checklist (Figure 1) to determine if virtual care was appropriate for the patient. If the patient was experiencing any of the listed high-acuity complaints (Figure 1), the family was directed to present for an in-person meeting. The authors found that virtual care could be a safe alternative to the traditional paediatric emergency department, with the ability to reduce the burden of in-person visits. Teleconsultations have already been known in the past as a helpful tool to facilitate emergency paediatric care [34,35]. Nevertheless, this is the first study on virtual emergency care with the pre-assessment in the form of an online checklist being successfully employed.

Returning to the input-throughput-output model, the throughput part seems to be a bottleneck in patient flow through paediatric emergency departments [36]. Compared to adult emergency departments, where the delay in the transfer of admitted patients limits the flow the most, operational inefficiency drives the flow in paediatric emergency departments [36,37]. One of the contributing factors could be long waits for the results of ordered tests. Ajmi et al. used the optimised workflow model to clarify that the delay on this level of operation was rather caused by missing alerts when results were available [38]. Such operational inefficiency might be effectively solved nowadays, for example, by the computerised whiteboard system described by Aronsky et al. [39]. The whiteboard system consists of a large, touch-sensitive monitor which displays an overview of all admitted patients and ongoing operations in the emergency department. The delay in the transfer of admitted patients, if identified as the limiting factor, could be easily overcome by implementing artificial intelligence to predict hospital admission at the time of triage and thus liberate a bed for a coming patient in advance [40,41,42].

Finally, any change toward accelerating the patient flow might be tested before its institution by a decision support system, which is based on a discrete-event simulation model and allows prediction of the impact of the intended change [43]. The successful application of lean thinking, i.e., focusing on value-adding steps and eliminating non-value-adding steps in every part of the input-throughput-output model, was also demonstrated [44,45] and can, by nature, serve as overall philosophy on how to increase the efficiency of paediatric emergency departments.

## 3. How to Optimise the Use of Structural Approach?

When under stress, clinicians are less able to recall remembered lists and are more likely to become fixated on a certain course of action and reluctant to change it, despite evidence that indicates a need for change [46,47]. In such an environment, it becomes easier to follow structured guidance. For simplicity, the presented look of the guidance may take the form of posters, flowcharts, checklists or even mnemonics, globally named cognitive aids. Cognitive aids lead to timely recognition and effective management of ongoing issues by improving communication [48,49], teamwork [49,50,51], and the safety culture [51,52,53]. Unsurprisingly, the use of cognitive aids is associated with a significant reduction in error rates [54].

Due to the listed evidence, many emergency departments have adopted the globally issued guidelines or at least used them to establish their own recommendations. However, the rate of guideline use is low [55,56,57], most likely because of the lack of applicability of the otherwise well-written and high-quality texts [58,59,60,61,62]. It goes together with the results of the recent cross-sectional survey from China, where the authors identified guidelines accessibility at the point of care and training of medical staff to better embrace guidelines as two key challenges in the way of successful guidelines implementation [63]. Considering such findings, more attention should be given during the process of guideline development, notably to designing their applicability.

If local health authorities intend to prepare new guidelines, an extensive literature review on the subject of guideline development and implementation from Kredo et al. [64] might render useful input. The course of new guideline development is also depicted in Figure 2. Once a gap in available guidelines is detected, there are well-credentialed guideline-development manuals from the World Health Organization [65], the Scottish Intercollegiate Guidelines Network [66], the National Institute for Health and Care Excellence [67], and the Australian National Health and Medical Research Council [68]. For simplicity, Schünemann et al. itemised all potentially relevant steps on the way of guideline development into the 18-point checklist [69]. Before the initiation of guideline development, available data in the intended area of study need to be gathered and graded according to their quality. For such synthesis of evidence, Grading of Recommendations Assessment, Development and Evaluation (GRADE) [70], or the Australian NHMRC approach, Formulating Recommendations Matrix (FORM) [71], have emerged. By the end of guideline development, the willingness of clinicians to use the guideline could be predicted by GuideLine Implementability Appraisal [72]. Regarding guideline presentation, physicians prefer the multilayered presentation format over the traditional narration [73]. In the multilayered format, the recommendations are clearly stated upfront, and the additional information pops out as collapsible boxes after clicking on the recommendation itself. The strength of the recommendation is communicated by colour coding, and a header describes the population to which the recommendation applies. A ‘user-friendly’ multilayer software tool for guideline presentation was issued by DECIDE consortium [74] and is available at http://www.decide-collaboration.eu/ (accessed on 20 March 2023).

Rather than preparing a new guideline, the local health authorities more often face up to an excess of guidelines on the same subject. The main role of the local health authorities then remains to choose the most appropriate guideline to be applied in the local health setting (Figure 2). To facilitate the selection, the Appraisal of Guideline ResEarch and Evaluation (The AGREE II) instrument [75,76] tests the quality of guidelines on 23 items; each scored 1–7 (Strongly Disagree through to Strongly Agree). The items are grouped into six domains (1—scope and purpose, 2—stakeholder involvement, 3—the rigour of development, 4—clarity of presentation, 5—applicability, and 6—editorial independence) and, by calculating the domain scores, the health care provider might easily identify strong and weak sides of the tested guideline. The successful application of AGREE II can be demonstrated by comparing seven high-quality international guidelines focused on the management of fever in children [77]. The authors selected two appraisers who underwent online training in the use of AGREE II and then independently assessed each of the seven guidelines. The calculated domain scores were summarised in the comparative table, and the authors could hereby depict the guideline issued by the British National Institute for Health and Care Excellence as the most recommendable one without the need for any further modification. Thus, any local health authority, furnished with AGREE II, might elegantly evaluate the quality of international guidelines and make a synthesis of the most suitable recommendations. As a simplified alternative to AGREE II, a quality checklist iCAHE was recently developed [78].

To foster guidelines implementation, cognitive aids should be involved since they improve adherence to guidelines [79,80]. The most frequent form of cognitive aids has become a checklist. However, despite its simplicity, instructiveness and proven positive impact on reducing mortality [81], its use may meet several obstacles, such as the operating theatre staff’s reluctance to perform a surgical safety checklist before every surgery [82]. Thus, the checklist must be designed to harmonise with the flow of emergency tasks and to minimally bother the clinical staff. Burian et al. recently provided a comprehensive and evidence-based manual for the development of medical checklists [83]. The traditional design of a checklist is a form of a step-by-step guide, most probably adopted from the field of aviation. However, Burian and his colleagues observed that clinicians do not follow a uniform and linear scenario when responding to critical events. Instead, many of them first solve the situation in their own way and only afterwards refer to a checklist for additional ideas or specific information (e.g., alternative drugs and their dosage) [84]. The emergency actions communicated by checklist should be therefore grouped into colour-coded blocks. This allows users to respond to an event already in progress by jumping directly to the needed block at any time. Some critical-event checklists for paediatric life-threatening events have already been designed and issued by the Society for Pediatric Anesthesia and are freely available on its webpage (https://www.pedsanesthesia.org/; accessed on 20 March 2023).

Training in the use of a checklist will increase the rate and success of its use [85]. For this purpose, the Society for Pediatric Anesthesia recommends that all checklist users, while being trained, should be familiarised with the structure and layout of the checklist as well as instructed on how the checklists are ordered (e.g., alphabetically) [86]. The other necessary parts of training are to specify who is involved in performing the checklist, to make users understand the goal of the checklist and expected actions for each event, to expose users to scenarios for which checklists are designed and to maintain proficiency by frequent reviewing. The translation of paper-based checklists into electronic ones to make them usable on smartphones or tablets might also render better outcomes [87,88,89]. Even a simple audio prompt was found to be helpful for improving adherence to guidelines [90,91] or to surgical safety checklists [92], most likely because it saves the visual attention of clinicians for other tasks.

The most frequently used structural approach is the ABCDE one. Initially introduced by Safar et al. [93], the ABCDE training was later proved as an important tool for improving survival following cardiac arrest [94]. A cognitive aid tool for the ABCDE approach was recently developed and validated in the simulation study [95]. Even though the ABCDE approach is considered a hallmark of emergency medicine, there is limited knowledge of how often and how completely it is applied to emergency patients. A recent study done by Olgers et al. showed that the ABCDE approach was performed more often and sooner after the admission of unstable patients with high triage levels [96]. While the triage level decreases, the ABCDE approach has been performed more sporadically despite the medical staff being well-trained in this approach with a high completeness score. The main reasons for omitting the ABCDE approach were that the patient seemed stable at first glance (clinical impression), and the vital signs done by the nurse did not indicate instability. It might look safe enough to use the clinical impression initially and only then to decide if an ABCDE approach is needed. However, since the ABCDE approach itself can be performed within 10 min in most patients, its application seems convenient even in stable patients.

The essential part of the structural approach in an emergency department is to have a properly formed resuscitation team. As suggested by Weng et al., the in-hospital resuscitation team should consist of 6 members: a team leader, a compressor, a recorder, and a member for intravenous access, for preparing medication and for keeping airway and ventilation [97]. The association between the establishment of the structured resuscitation team and the increased rate of return of spontaneous circulation during cardiopulmonary resuscitation is well-documented [97]. However, delay in identifying team roles still represents a substantial part of system errors in cardiopulmonary resuscitation [98]. The team can be formed from the members of the emergency department for the purpose of providing immediate cardiopulmonary resuscitation to admitted patients. It is therefore recommended that the members of the emergency department should meet at the beginning of each shift for introductions and allocation of roles in the resuscitation team [99].

## 4. How to Rationalise the Use of Imaging Methods?

Ultrasound can be of diagnostic help in multiple emergency settings. Abdolrazaghnejad et al. brought a comprehensive summary of the ultrasound protocols used in emergency medicine and proved that ultrasound decreases the time needed for diagnosis and treatment [100]. We rank among the most established point-of-care ultrasound (POCUS) protocols Extended Focused Assessment with Sonography in Trauma (eFAST), Bedside Lung Ultrasound in Emergency (BLUE), Rapid Assessment of Dyspnea with Ultrasound (RADiUS) for dyspnoea, Rapid Ultrasound in Shock (RUSH) for shock and Focused Echocardiography in Emergency Life support (FEEL) for cardiac arrest [101]. These protocols are mostly standardised for the adult population. Nevertheless, they can also be used in children, bearing in mind anatomical and physiological differences between adult and child patients.

On the other hand, some studies demonstrated either no benefit [102] or deterioration [103] in the outcome if ultrasound was added to the initial emergency management. The evidence brings us to the conclusion that routine ultrasound use in every patient (even with an evident diagnosis or in a state which is treatable without having the exact diagnosis) does not yield benefits. The use of ultrasound should thus be targeted to answer specific clinical questions (e.g., use of BLUE protocol in a patient with the clinical sight of respiratory failure to help us differentiate the presence of pulmonary oedema, pneumothorax, or other). An overview of the emergent cases where ultrasound was thought to be able to improve outcomes was recently done by Goldsmith et al. [104].

Regarding emergency ultrasound education, the recommendation for ultrasound training was issued by the American College of Emergency Physicians [105] and also by Vieira et al. as consensus educational guidelines [106]. Blehar et al. determined a minimum of 50 examinations that any learner must perform to reach a performance level comparable to expert sonographers for both image acquisition and interpretation [107]. To experience a sufficient number of examinations, simulations and multimedia resources might be involved [108]. Implementing ultrasound training into medical school curricula may also reduce educational burdens for emergency physicians [109,110].

Quantitative assessment of ultrasound images, provided automatically by artificial intelligence, remains nowadays a debatable topic [111]. It was shown that ultrasound examination, augmented by artificial intelligence, increased accuracy and efficiency for diagnosing pneumonia by lung ultrasound [112], interpreting echocardiogram [113] or detecting and predicting the prognosis of cancer disease [114]. Ultrasound can also be used for the diagnosis of long bone fractures. Very promising is the POCUS diagnosis of paediatric forearm fractures with a pooled sensitivity of 93.1% and specificity of 92.9% [115]. POCUS can also be used in the control of close reduction of fractures in the emergency unit as a quick and sensitive diagnostic method. In addition to ultrasound, artificial intelligence can be useful for fracture diagnosis on radiographs particularly if the specific type of specialist is not available in the hospital setting [116].

Talking about imaging methods in the paediatric emergency department, the right indication of head imaging in minor head trauma belongs to a tricky task for every paediatric emergency physician. Head injury in children has been getting more common in the last decade, fortunately, with a low incidence of severe cases requiring neurosurgical or other therapeutic intervention. However, it still represents one of the most common causes of disability and death at a young age [117]. After clinical examination, the standard diagnostic method for head trauma is a computed tomography (CT) scan. Even though X-ray is more accessible to emergency physicians, its suitability in minor head trauma is questioned since it gives us no information about intracranial changes [118]. An X-ray can identify a skull fracture not apparent by clinical examination. However, up to 50% of intracranial trauma can be present without a skull fracture. With a high incidence of minor head trauma, often repeated in the same patient, a CT scan might be problematic because of unnecessary and high radiation exposure. In comparison with adults, children are more sensitive to radiation with longer life expectancy than adults. Moreover, if CT settings are not adjusted for children’s body, they can get an unnecessarily higher dose of radiation with the increased risk of malignant disease as tumors or leukemia [119]. For a cumulative dose of 50 to 60 milligray to the head (equivalent of two to three CT scans), a threefold increase in the risk of brain tumors was reported.

Taken together, it is of high importance to have clinical decision rules to select high-risk patients with severe head trauma. Three algorithms were validated for this purpose: Pediatric Emergency Care Applied Research Network (PECARN), Children’s Head Injury Algorithm for Prediction of Clinically Important Events (CHALICE), and Canadian Assessment of Tomography for Childhood Head Injury (CATCH),see Figure 3 [120]. According to prospective cohort studies [121,122], PECARN showed the highest sensitivity in comparison with two other decision rules. Schonfeld et al. proved that children in a very low-risk group for traumatic brain injury, according to PECARD, could safely avoid a CT scan with a very low risk of significant head injury [123]. In general, all three decision rules are based on similar premises and are summarised in Figure 3.

A very promising method in the detection of paediatric skull fractures is becoming ultrasound. According to some studies, bedside emergency ultrasound performs with 100% sensitivity and 95% specificity when compared to CT scans for the diagnosis of skull fractures [124]. This can significantly reduce excessive radiation exposure in children after minor head trauma. Thanks to the simplicity of this examination, the emergency physician does not need to have great experience to get an accurate image.

## 5. Conclusions

The increasing attendance of paediatric emergency departments evoked the necessity for care optimisation in an effort to guarantee the patient’s safety. Hereby, we presented the compilation of several strategies whose single use in the emergency setting somehow led to outcome improvement. The idea of potentiating the benefits of the proposed improvements by their combination remains to be elucidated. The review serves not only as a blueprint for modernising paediatric emergency departments but also as a pool of useful literature which can be suitable in the paediatric emergency field.

## Figures and Tables

**Figure 1 children-10-00741-f001:**
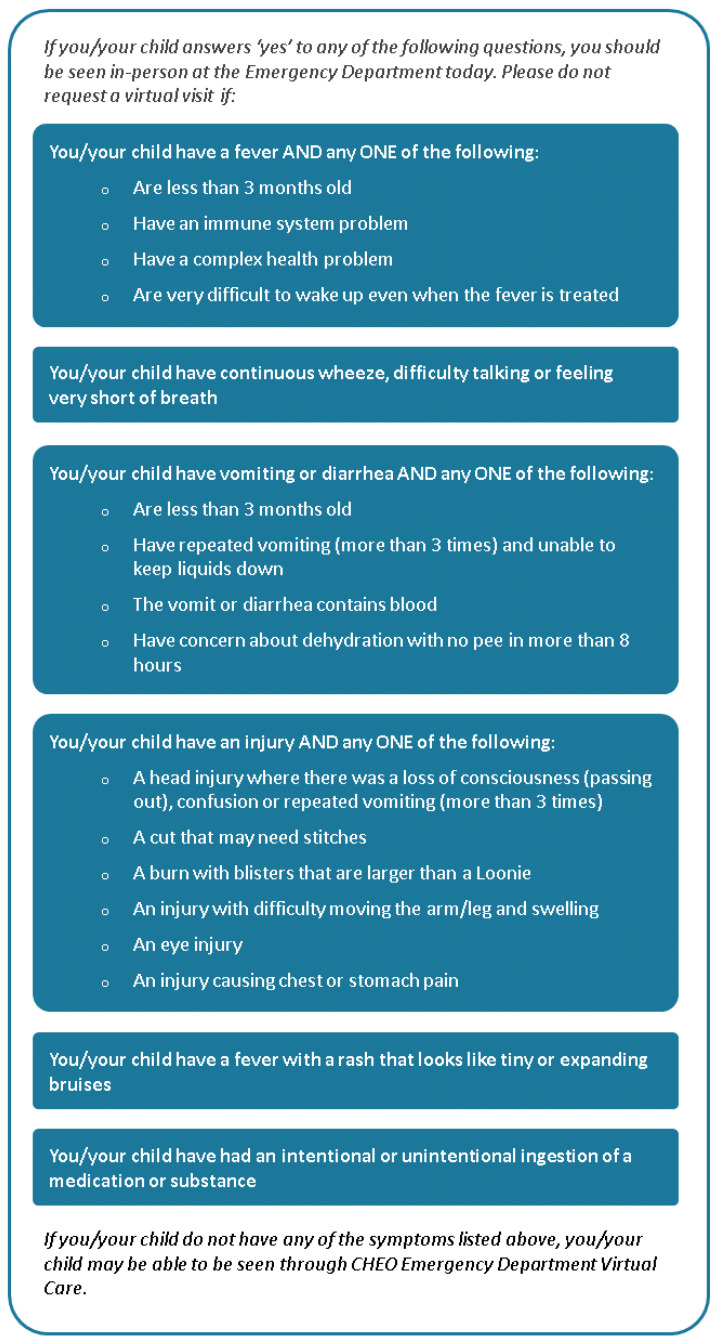
Screening checklist to determine if virtual care is appropriate for the paediatric patient. Adopted from Reid et al. [33]. The checklist is done online prior to meeting an emergency physician. If the patient experiences none of the stated high-acuity complaints, the required care and follow-up can be provided virtually.

**Figure 2 children-10-00741-f002:**
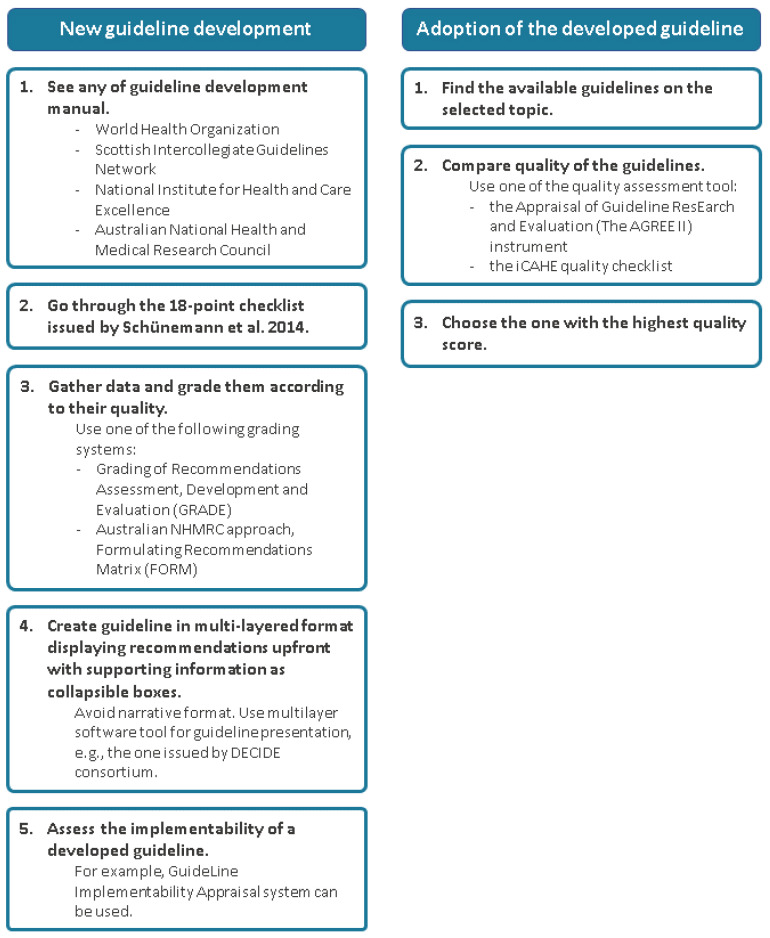
The possible step-by-step manual for the new guideline development or for the adoption of already-developed guidelines.

**Figure 3 children-10-00741-f003:**
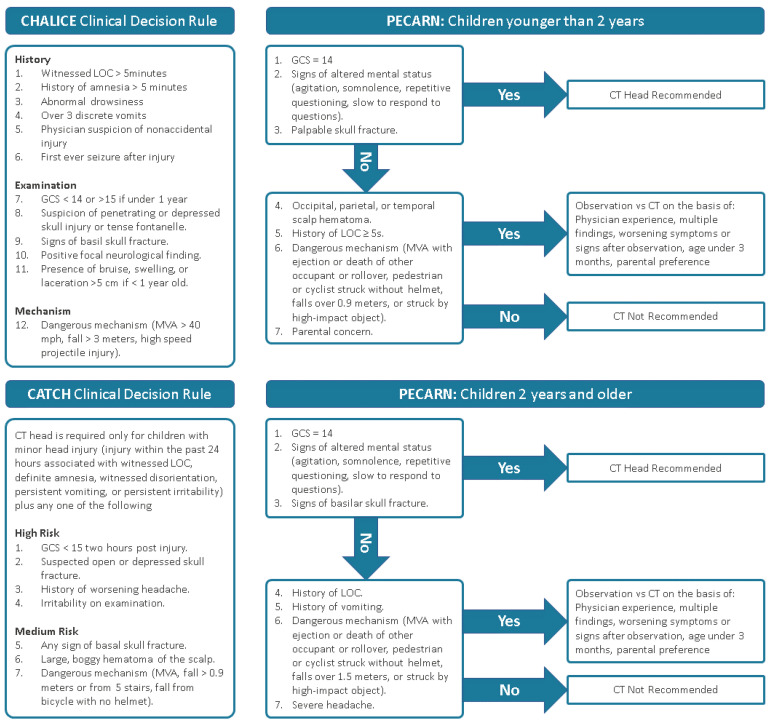
Summary of Clinical Decision Rules (CHALICE, CATCH and PECARN) to identify candidates with minor head trauma for head CT scan. Adopted from McGraw and Way [120]: LOS—loss of consciousness, MVA—motor vehicle accident, mph—miles per hour.

**Table 1 children-10-00741-t001:** Triage system characteristics. Table is adapted from Hinson et al. [15].

Triage System	CTAS	ESI	MTS	ATS	SATS
Stated objective	Provide patients with timely care	Prioritise patients by immediacy of care needs and resource	Rapidly assess a patient and assign a priority based on clinical need	Ensure patients are treated in order of clinical urgency and allocate patients to the most appropriate treatment area	Prioritise patients based on medical urgency in contexts where there is a mismatch between demand and capacity
Recommended time to physician contact, min	1: immediate	1: immediate	Red: immediate	1: immediate	Red: immediate
	2: ≤15	2: ≤15	Orange: ≤10	2: ≤15	Orange: ≤10
	3: ≤30	3: none	Yellow: ≤60	3: ≤30	Yellow: ≤60
	4: ≤60	4: none	Green: ≤120	4: ≤60	Green: ≤120
	5: ≤120	5: none	>Blue: ≤240	5: ≤120	Blue: ≤240
Discriminators					
Clinical	Yes	No	Yes	Yes	Yes
Vital signs	Yes	Yes	Yes	Yes	Yes
Pain score	Yes (10-point)	Yes (visual scale)	Yes (3-point)	No	Yes (4-point)
Resource use	No	Yes	No	No	No
Paediatrics	Separate version	Separate vital sign differentiators	Considered within algorithm	Considered within algorithm	Separate flowchart

## Data Availability

This review does not report any study data.

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
