# Peer review of "Modern Paediatric Emergency Department: Potential Improvements in Light of New Evidence"

_children, 2023, doi:10.3390/children10040741_

Round 1
Reviewer 1 Report
I would like to thank the editor for the opportunity to review this manuscript relating to a field of interest to me. the text is well written but I'm not a native speaker so I can't suggest any particular changes to English, and I'll leave the task of correcting the language to the editor.
I suggest some minor revisions.
Paragraph 2: I suggest talking about overcrowding first and then triage schemes as the text can be confusing.
paragraph 3: the paragraph is very long so it might be useful to try to shorten it and above all divide it into subparagraphs to make it more readable.
paragraph 4: it could be useful to divide the text into two sub-paragraphs, one dedicated to POCUS, further emphasizing the type of training necessary for this practice which is now necessary in all pediatric emergency departments, and one dedicated to CT; I suggest deleting the comment relating to the use of radiography as it is now an obsolete practice as also reported in the algorithms (PECARN) mentioned in the manuscript.
Reviewer 2 Report
The topic of this manusript is really interesting and of importance to all physicians who treat emergency patients. The authors presented what is so far known about the problems in this area. They also managed to clerly present possible ways to improve and optimize workflow in Pediatric EDs.
Author Response
We thank the reviewer for their valuable and supportive comment!